# Mesenchymal Stem Cells in Myelodysplastic Syndromes and Leukaemia

**DOI:** 10.3390/biomedicines12081677

**Published:** 2024-07-26

**Authors:** Ilayda Eroz, Prabneet Kaur Kakkar, Renal Antoinette Lazar, Jehan El-Jawhari

**Affiliations:** 1Biosciences Department, School of Science and Technology, Nottingham Trent University, Nottingham NG11 8NS, UKprabneet.kakkar2023@my.ntu.ac.uk (P.K.K.); renal.lazar2023@my.ntu.ac.uk (R.A.L.); 2Clinical Pathology Department, Faculty of Medicine, Mansoura University, Mansoura 35516, Egypt

**Keywords:** mesenchymal stem cells, leukaemia, bone marrow microenvironment, myelodysplastic syndromes

## Abstract

Mesenchymal stem cells (MSCs) are one of the main residents in the bone marrow (BM) and have an essential role in the regulation of haematopoietic stem cell (HSC) differentiation and proliferation. Myelodysplastic syndromes (MDSs) are a group of myeloid disorders impacting haematopoietic stem and progenitor cells (HSCPs) that are characterised by BM failure, ineffective haematopoiesis, cytopenia, and a high risk of transformation through the expansion of MDS clones together with additional genetic defects. It has been indicated that MSCs play anti-tumorigenic roles such as in cell cycle arrest and pro-tumorigenic roles including the induction of metastasis in MDS and leukaemia. Growing evidence has shown that MSCs have impaired functions in MDS, such as decreased proliferation capacity, differentiation ability, haematopoiesis support, and immunomodulation function and increased inflammatory alterations within the BM through some intracellular pathways such as Notch and Wnt and extracellular modulators abnormally secreted by MSCs, including increased expression of inflammatory factors and decreased expression of haematopoietic factors, contributing to the development and progression of MDSs. Therefore, MSCs can be targeted for the treatment of MDSs and leukaemia. However, it remains unclear what drives MSCs to behave abnormally. In this review, dysregulations in MSCs and their contributions to myeloid haematological malignancies will be discussed.

## 1. Introduction

The International Society for Cell & Gene Therapy Mesenchymal Stromal Cell (ISCT MSC) committee defines mesenchymal stem cells (MSCs) as multipotent progenitor cells that possess the ability to self-renew and differentiate into various cell types of the mesodermal lineage, including osteocytes, chondrocytes, adipocytes, and myoblasts, as well as cells of embryonic lineages [1]. Due to their capacity for multipotent differentiation (Figure 1), MSCs have immense value in regenerative medicine and tissue engineering applications. MSCs are also characterised by the positive expression of CD105, CD90, and CD73, as well as negative expression of endothelial and haematopoietic markers, such as CD14, CD34, CD45, human leukocyte antigen-DR isotype (HLA-DR), and CD19 in humans [2]. The adherence to culture surfaces is another characteristic of in vitro-expanded MSCs. MSCs can be collected from various body sources, including bone marrow (BM), placenta, adipose tissue, muscles, umbilical cord, amniotic fluid, menstrual blood, cord blood, dental pulp, dermal tissue, and urine [3].

MSCs also play a vital role in regulating and maintaining the properties of haematopoietic stem cells (HSCs) [4]. The interaction between HSCs and MSCs prevents the differentiation of HSC into B-cells and dendritic cells (DCs) and protects them from apoptosis, promoting HSC stemness and self-renewal [5]. Furthermore, MSCs have an immunosuppressive effect that can suppress various immune cells like macrophages, natural killer (NK) cells, B cells, T cells, and dendritic cells, which can be useful in treating autoimmune diseases and conditions involving excessive inflammation [6].

MSCs can produce several bioactive molecules [7], including growth factors (transforming growth factor beta (TGF-β), epidermal growth factor (EGF), granulocyte-macrophage colony-stimulating factor (GM-CSF)), adhesion molecules (activated leukocyte cell adhesion molecule (ALCAM), intercellular adhesion molecule-1 (ICAM-1)), cytokines (e.g., interleukin-IL-1α, IL-1β, IL-6, TNF-α), immunomodulatory molecules (prostaglandin E2 (PGE2), human leukocyte antigen (HLA-G)) [8], and angiogenic factors (vascular endothelial factor (VEGF), platelet-derived growth factor (PDGF)) [9] that are responsible for the paracrine effect of MSCs on neighbouring cells. Due to the paracrine effect, MSCs can influence the behaviour of surrounding cells, making them valuable for modulating tissue regeneration and immune and inflammatory responses [10].

The BM is made up of different cell types, including endothelial cells, MSCs, pericytes, adipocytes, osteoblasts, HSCs, and haematopoietic cells, such as macrophages, osteoclasts, neutrophils, and regulatory T cells [11]. BM-resident cells like HSCs, endothelial cells, and cells derived from MSCs, in conjunction with the extracellular matrix (ECM), create a highly specialised microenvironment responsible for regulating the formation of mature haematopoietic cells and their proper functions [12]. The dysregulation of BM niche cells interferes with stem cell behaviour, resulting in malignant transformation [13].

## 2. The Role of MSCs in Pre-Leukaemia Myelodysplastic Pathogenesis 

Myelodysplastic syndromes (MDSs) comprise a heterogeneous group of clonal haematopoietic disorders that affect haematopoietic stem and progenitor cells. These disorders are characterised by inefficient haematopoiesis, BM failure, and peripheral blood cytopenia with a high risk of progression to acute leukaemia [14]. Studies have shown that there are various mutations in somatic genes present in HSCs [15], particularly those involved in RNA-splicing (SBDS and DICER1) [16], signal transduction, DNA methylation, chromatin modification, cohesion regulation, transcription factors (p53), cytogenetic abnormalities [17].

In 2022, WHO divided MDS subtypes based on BM cellularity, the presence of mutations, ring sideroblasts, concomitant pathologies such as lymphoma, infection, and metastasis, dysplasia, the level of BM fibrosis, and BM karyotype by adopting haematologic, morphologic, cytogenetic, and molecular genetic approaches (Table 1) [18].

Finally, The International Prognostic Scoring System (IPSS), a system used for the assessment of prognosis in MDS patients, classifies MDSs based on the percentage of BM blasts, karyotype, and cytopenia(s) and scores patients as below (Table 2).

Some in vivo models have been developed for MDS studies using xenotransplant mice by engrafting immortalised human cell lines derived from MDS patients. Moreover, genetically engineered mouse models have been established by reverse transcription BM transduction/transplantation, including harvesting murine BM nucleated cells with a retroviral construct that can express the gene of interest. There is another type of genetically engineered mouse model made via gene targeting to attain mouse embryonic stem cells positioned to form haematopoietic mouse cells that have human MDS properties [19]. Similarly, MDSs can be developed in rats exposed to chemical agents like DMBA. Moreover, a genetically engineered model of zebrafish can be utilised for MDS studies through the disruption of the Tet2 catalytic domain, whose loss-of-function mutations are common in MDSs [20].

To study MDSs in vitro, some cell lines derived from MDS patients such as MDS92, MDS-L, M-TAT, and TER-3 and cell lines obtained from leukaemia developed after MDS including MDS-KZ and SKM-I can be utilised. Moreover, the induced pluripotent stem cells obtained from somatic cells of MDS patients are able to differentiate into different haematopoietic lineages and can be reverted to modelling lower-stage disease via gene corrections [21]. Lastly, co-culture systems can be created with MDS cells and stromal and/or haematopoietic cells to investigate cellular crosstalk [22].

### 2.1. Impaired Morphology and Immunophenotype of MSCs in MDS

Cultured MDS BMSCs have been found to exhibit an irregular morphology. For example, one study found that ex vivo expanded patient-derived BMSCs exhibited thicker and granular morphology [23]. However, several studies suggest that variations in experimental methodologies and patient heterogeneity may contribute to conflicting findings on this topic. Specifically, some studies [24] have failed to detect morphological abnormalities in BMSCs in MDS patients in comparison to their healthy counterparts [25]. Despite these discrepancies, most studies [26] agree that patient BMSCs do not differ significantly from normal MSCs in terms of immunophenotype [27]. However, some studies report reduced expression of several key markers, including CD34, CD90, and CD45, on the surfaces of patient BMSCs [28].

### 2.2. Cytogenetic and Genetic Abnormalities of MSCs in MDS

Patients with de novo MDSs frequently have clonal cytogenetic abnormalities found in their haematopoietic cells. On the other hand, there is inconsistent information on genetic abnormalities in MDS-BMSCs. While some research [29] has demonstrated that patient ex vivo grown BMSCs display chromosomal abnormalities such as 5q deletion, other publications suggest that cytogenetic analysis of these cells is normal. Notably, the clonal chromosomal aberrations seen in MDS-BMSCs consistently differ from those found in the patient’s haematopoietic cells, indicating that the two cell types are not produced from the same clone. This idea is further supported by the discovery that the mesenchymal compartment of BM-mononuclear cells from MDS patients did not include any mutations of epigenetic or spliceosomal genes, including TET2, DNMT3A, SRSF2, and SF3B1. Notably, research on the detection of chromosomal abnormalities in BMSCs from MDS patients is currently ongoing. MDS-BMSCs displayed a higher mutational burden and different mutational signatures compared to healthy BMSCs according to a recent study that used exome sequencing to examine the prevalence of clonal mutations in ex vivo grown BMSCs from both patients and healthy donors. However, primary stroma cells from the same patients did not exhibit the extremely recurrent mutations found during culture. These results imply that the stroma compartment of MDS patients does not contain any evidence of clonal mutations [30].

Cancer cells can accumulate epigenetic changes that support their progression [31]. Abnormal changes in epigenetics occur not only in cancer cells but also in the microenvironment. It has been found that epigenetic alterations can affect BMME, contributing to MDSs. For instance, the binding sites of some transcription factors play a role in haematopoiesis, like MDSs. For example, the binding sites of some transcription factors play a role in haematopoiesis, for example, NF1 and Myb were methylated in MDS stroma, indicating that MDS stromal cells have abnormal hypermethylation that can impact the Wnt/B-catenin pathway [32]. Moreover, in MDSs, there are many mutated genes involved in DNA repair, RAS signalling, and chromatin modification like ATM, PTPN11, and EZH2 located in the germline, leading to a predisposition to AML and MDSs [33].

Studies have demonstrated that, in comparison to healthy MSCs, MDS-associated MSCs exhibit increased levels of genotoxic stress indicators, such as the frequency of phosphorylation of yH2AX foci [34] and replication protein A (RPA) [35]. The staining results of yH2AX in MDS patients were found to be correlated with the frequency of mutations. Mutations in MSCs activated by β-catenin are linked to the emergence of myelodysplastic alterations and common genetic changes in AML in MDSs [36]. When MSCs’ β-catenin is activated, Jagged1 is expressed, which causes haematopoietic stem and progenitor cells (HSPCs) to activate Notch signalling, accelerating the development of leukaemia. Overall, because of these anomalies, BMSCs may accelerate the course of the disease.

### 2.3. Abnormal Haematopoietic Microenvironment Induced by MSCs in MDSs

MDS-BMSCs have demonstrated the capability of contributing to the abnormal haematopoietic microenvironment but with no clear conclusions. Some studies suggest that patient BMSCs can sustain the growth of both leukaemic/autologous HSPCs and HSPCs derived from healthy individuals [37]. Conversely, other studies indicate that MDS-BMSCs have a diminished capacity to support normal HSCPs [38]. These observations are attributed to the impaired expression of niche-derived molecules that are known to influence haematopoiesis, such as angiopoietin (ANGPT), osteopontin, Jagged-1, hepatocyte growth factor (HGF), kit ligand, C-X-C motif chemokine ligand 12 (CXCL12), thrombopoietin (TPO), and insulin growth factor binding protein 2 (IGFBP2), as well as TGFβ1 [39]. 

Notably, Jagged-1, which is involved in the Notch signalling pathway, is crucial for the differentiation of HSCs. Impaired Jagged 1 expression might lead to the disruption of this process [40]. Additionally, the upregulation of inflammatory factors and inhibitors of haematopoiesis due to the activation of NF-kB, such as IL-8, IL-6, and C-C motif chemokine ligand (CCL3), has been demonstrated to attenuate HSPC numbers and function ex vivo in the CD271+ BMSCs from lower-risk MDS (LR-MDS) [41]. Moreover, the Sbds deletion observed in osterix+ MDS-MSCs promotes genotoxic stress in HSPCs through inflammatory p53-S100A8/9-TLR signalling, leading to MDS progression [42]. Another study has indicated that when healthy haematopoietic cells are exposed to MDS-MSCs, they will have a toxic effect on CD34+ HSPCs [43]. Furthermore, MSCs derived from patients with both early-stage and late-stage MDSs exhibited decreased expression of angiopoietin/angiopoietin-like, a protein that regulates HSPC quiescence [44]. 

Exosomes secreted by ex vivo expanded BMSCs from LR-MDS patients have been shown to have a different miRNA cargo than their normal counterparts. Patient BMSC-derived exosomes are incorporated into CD34+ cells from healthy donors, change their gene expression through miRNA transfer (such as miR-15a and miR-10a), and increase their clonogenic potential and viability [45]. Also, it was found that the co-culturing of megakaryocytes with MSCs induces alterations in LAT and Rap1b gene expression, resulting in the generation of platelets that exhibit low basal activation levels, suggesting its contribution to the MDS development [46]. Figure 2 illustrates the relationship between MSCs and haematopoietic cells in the healthy and MDS/AML BMME. 

### 2.4. Immunomodulatory Dysfunction by MSCs in MDSs

BMSCs in MDS patients have exhibited a defect in their ability to inhibit T-cell proliferation and activation [47]. Some studies have also demonstrated that MDS-BMSCs have an impaired capacity to inhibit T-cell proliferation when stimulated in mixed-lymphocyte reactions [27]. Furthermore, one study found that MDS-BMSCs were less effective in inhibiting T-cell proliferation but had a similar capacity to induce regulatory T-cells compared to normal BMSCs. High-risk MDS MSCs (HR-MDS) showed a lower inhibition effect on T cell apoptosis compared to low-risk MDS MSCs (LR-MDS MSCs). Even though both HR-MDS MSCs and LR-MDS MSCs suppressed T cell proliferation, the immunosuppressive impact of LR-MDS MSCs on the proliferation of T cells was less than that of HR-MDS MSCs. Moreover, a role for the TGF-β1 produced by MDS-MSCs was suggested in the generation of Tregs [48]. Interestingly, it has been reported that the BMSCs obtained from MDS patients can induce normal monocytes to acquire the properties of myeloid-derived suppressor cells, eventually leading to the downregulation of NK and T cell function [49]. The impaired ability of BMSCs to inhibit the activation and proliferation of T cells may contribute to immune dysregulation, leading to the progression of MDS and potentially contributing to the evolution of MDS to AML.

### 2.5. Cytokine Dysregulation Mediated by MSCs in MDSs

MDS-MSCs demonstrate signs of ongoing stromal stimulation and response to an inflammatory environment, which is accompanied by increased expression of genes associated with fibrosis (ADAMTS4, SPARC, and LOXL2), a feature of MDSs [44]. A study has also highlighted that a gene set, including cytokine–cytokine receptor interaction, is significantly enriched in MDS MSCs. The CXCR4/CXCL12 signalling pathway sustains haematopoiesis in the BM through MSCs, as well as a reduction in the expression of some haematopoietic factors, such as stem cell factor (SCF), insulin-like growth factor 1 (IGF1), and TPO [50], leading to MDSs by an impairment in the normal process of blood cell formation and the growth and differentiation of HSCs and progenitor cells that contributes to the ineffective haematopoiesis characteristic of MDS and the production of dysfunctional and abnormal blood cells. In more detail, SCF is known to be involved in maintaining the HSC niche within BM to provide a supportive environment for haematopoiesis; for this reason, a decrease in SCF expression might disrupt the integrity of the niche, affecting the interactions between HSCs and the BMME and the self-renewal/differentiation of HSCs, contributing to MDS pathogenesis. TPO is a key regulator of platelet production, stimulating the differentiation of megakaryocytes and subsequent platelet release. A reduction in TPO expression might lead to thrombocytopenia characterised by low platelet counts, which is a common feature in MDS and can contribute to bleeding tendencies [51]. 

A decrease in IGF1 playing a role in promoting cell survival and growth might affect growth signalling pathways, potentially affecting the survival and proliferation of haematopoietic cells. Dysregulation of growth signalling is implicated in the abnormal expansion of malignant clones seen in MDSs [52].

### 2.6. Altered Growth Kinetics and Elevated Cellular Senescence of MSCs in MDSs

Studies have shown that MDS-derived BMSCs have impaired growth potential that is associated with a decrease in the expression of CD44 and CD49e [53]. The presence of CD49e is crucial for facilitating cellular attachment to the ECM [54]. CD44 is essential for the homing and adhesion of haematopoietic progenitor cells (HPCs) to MSCs, facilitating the establishment of a supportive microenvironment for haematopoiesis [55].

The lower proliferative capacity of MDS-BMSCs could also be attributed to the downregulation of the canonical Wnt signalling pathway and the upregulation of the non-canonical pathway. A recent study confirmed the downregulation of the canonical Wnt signalling pathway in BMSCs derived from MDS patients [56]. Additionally, another study that used the samples of patients with different MDS subtypes such as RCMD, RAEB I, RAEB II, del5q, CMML, and MDS-U has shown that MSCs from MDS patients are more susceptible to cellular senescence. 

The increased senescence and impaired growth kinetics may affect the ability of MSCs to maintain a supportive microenvironment for HSCs and consequently hinder normal haematopoiesis. Findings have demonstrated a decrease in the haematopoietic expansion of CD34+ cells and colony-forming cell (CFC) potential following the co-culture of healthy HSPCs and HR-MDS-MSCs [41]. On the other hand, a study conducted by Rathnayake et al. analysed patients with different de novo MDS subtypes and found that culture-expanded MDS MSCs across all MDS types and healthy controls exhibited thin spindle and fibroblast-like cellular morphology [57]. Unlike other studies, MDS MSC doubling times were not remarkably different from that of controls or among the subtypes. These discrepancies might be due to differences in the culture conditions or experimental methodologies, different pathophysiologies for MDS classes, and heterogeneity of MDS biology [57].

### 2.7. Reduced Osteogenic Differentiation Caused by MSCs in MDS

Studies have shown that MDS-MSCs do not exhibit significant differences in adipogenic or chondrogenic differentiation, but their potential for osteogenic differentiation is remarkably reduced. This reduction in osteogenic differentiation is due to a decrease in the levels of microRNAs (miRNAs) involved in the early differentiation process toward osteoblasts, such as Osterix, a transcription factor, and OC, a marker of mature osteoblasts. A deletion of Dicer1, a protein involved in the processing of miRNAs, has been linked to the impaired osteogenic differentiation of MSCs related to MDS evolution in mice. Dicer deletion has been shown to inhibit adipogenic and osteogenic differentiation and decrease the support of healthy CD34+ cells. The same study also showed that all MDS subtypes exhibited osteogenic differentiation capacity, which is evident from decreased Runx2, a molecule essential for skeletal morphogenesis and osteoblastic differentiation, and alkaline phosphatases (ALP), an indicator of the new bone formation and the presence of osteoblast cells. Interestingly, when Dicer1 is overexpressed, these impaired functions are reversed [58].

Numerous studies have demonstrated that MDS-MSCs exhibit impaired PI3K/AKT and Wnt/β-catenin signalling pathways, inhibiting osteogenic proliferation and thereby contributing to the pathogenesis of MDSs [59]. A study on the transiliac bone biopsies obtained from MDS patients demonstrated abnormalities in bone remodelling that was characterised by reduced numbers of osteoclasts and osteoblasts, as well as decreased bone formation, as evidenced by the diminished mineral apposition rate [60]. A report consistent with these findings has also demonstrated bone loss in MDS patients [61]. Altogether, the reduced osteogenic differentiation capacity of MSCs can significantly impair their ability to support HSPCs, thereby leading to ineffective haematopoiesis in MDSs.

## 3. Role of MSCs in Leukaemia Pathogenesis

Leukaemia is a group of malignant tumours characterised by the abnormal and uncontrolled growth of cells that originate from haematopoietic precursor cells in the BM. Therefore, the BMME plays a crucial role in the development of leukaemia.

The classification of leukaemia is an integral component of the WHO classification of tumours, based on evidence used for different purposes, such as research and diagnosis, monitoring public health, and registries of cancer on a global scale [62]. The classification places a priority on categorising tumour types based on the identification of defining genetic abnormalities whenever possible and includes emerging entities as subtypes of the disease under a framework of other specified genetic alterations. Myeloid leukaemia encompasses the subtypes AML and chronic myeloid leukaemia (CML). Lymphoid leukaemia arises from the abnormal growth of lymphoid cells, which play a role in producing immune cells known as lymphocytes. It includes subtypes such as acute lymphoblastic leukaemia (ALL) and chronic lymphocytic leukaemia (CLL). Histiocytic/dendritic cell leukaemia is a rare type of leukaemia that originates from the abnormal growth of histiocytic or dendritic cells and is part of the immune system. In certain cases, MDSs can progress to leukaemia, known as MDS-related leukaemia or AML myelodysplasia-related (AML-MR).

MSCs located in the BMME are believed to be important in leukaemia development [63]. It is worth mentioning that MSCs exhibit both anti-tumorigenic and pro-tumorigenic effects in leukaemia. On one hand, MSCs, irrespective of their origin, can impede tumour growth by suppressing tumour cell proliferation, which is achieved by arresting the tumour cell cycle, inhibiting angiogenesis and decreasing vascular density and their paracrine action, including their ability to secrete extracellular vesicles that can exert an anti-proliferative impact on leukaemic cells. On the other hand, they can facilitate tumour growth by inhibiting tumour cell apoptosis [64]. 

### 3.1. Pro-Tumorigenic Effects of MSCs in Leukaemia

Numerous studies have demonstrated that MSCs can protect leukaemic cells and enhance their survival through multiple mechanisms. 

In a study conducted by Manabe et al., it was discovered that MSCs possess an anti-apoptotic activity in acute lymphoblastic leukaemia via the secretion of cytokines and soluble factors [65]. Another study found that MSCs were able to reduce apoptosis in BV173 CML cells. Upon contact with MSCs, leukaemic cells were found to be in a resting state (G0/G1) and a downregulation of cyclin D2 was observed. This downregulation is believed to preserve the proliferative capacity of the leukaemic cells, ultimately improving their chances of survival and self-renewal ability [66]. Another study revealed that MSCs can facilitate the growth of ALL cells in an in vivo model by increasing luciferase activity [67]. In addition, a study conducted by Naderi et al. using demonstrated that BMSCs can protect BCP-ALL cells from p53 accumulation and apoptosis. This is achieved through the production of PGE2 and activation of the cAMP-PKA signalling pathway, ultimately inducing leukaemogenesis [68].

MSCs directly contribute to the density of the tumour vasculature by differentiating into endothelial cells [69] or pericytes involved in tumour stabilisation [70]. Additionally, MSCs indirectly support tumour vasculature by secreting pro-angiogenic factors, including VEGF and other pro-angiogenic cytokines, such as fibroblast growth factor (FGF), ANGPT-1, and IL-6, which are necessary for the angiogenic activity of VEGF [71,72,73]. AML-MSCs have been found to have high levels of VEGFA and IL-6, which contribute to tumour-supportive angiogenesis [74,75]. Similarly, CLL patients have an increased amount of PDGF and VEGF. In MSCs, PDGF selectively activates the PDGF receptor (PDGFR), which induces MSC VEGF production through a PI3K-dependent mechanism, leading to neovascularisation and disease progression [76].

MSCs can lead to tumour growth by their immunomodulation effect. Granulocytic myeloid-derived suppressor cells primed with CML-MSCs exhibited an increase in immunomodulatory factors, including TNF-α, IL-1β, Arginase-1, IL-6, and cyclooxygenase 2 (COX-2), thereby supporting the contribution of MSCs to tumorigenesis [77]. Another study revealed that MSC stimulation in ALL led to the upregulation of the C-C motif chemokine ligand 2 (CCL2) and IL-8 inflammatory cytokines, which increased the adhesion support of MSCs for ALL cells, ultimately enhancing the proliferation and survival of ALL cells [78]. Additionally, MSCs were found to cause the dysfunction of T cells and the proliferation of Tregs in HR-MDS in CML patients and they were strongly associated with changes in MSCs [79,80]. Furthermore, CML-MSCs could activate regulatory DCs, leading to the inhibition of T cell function or the promotion of Treg proliferation, hence indirectly promoting immune escape in CML [81].

MSCs have been found to enhance the stemness of cells in blood cancers through the activation of the Bruton tyrosine kinase signalling pathway [82]. However, further research is needed to understand this mechanism fully.

MSCs can exert their pro-tumorigenic effects by inducing metastasis. The chemokines produced by MSCs can affect the migration of leukaemic cells and increase their metastatic potential [83]. Moreover, Activin A, which is a protein that can elevate the migratory capacity of leukaemic cells in response to the chemoattractant CXCL12 while suppressing the migration of healthy CD34+ cells, was found to be highly expressed by MSCs in the BMs of paediatric B-ALL patients compared to healthy counterparts, giving B-ALL cells a greater chance to metastasise, as observed in a xenograft mouse model and in vitro [84].

Of note, studies are ongoing to reverse the drug resistance contributed by the tumorigenic role of MSCs in leukaemia. For instance, Vianello et al. reported that MSCs cause leukaemia drug resistance by protecting leukaemia cells from imatinib-induced apoptosis through the CXCR4/CXCL12 axis [85]. Another study determined that MSCs have an anti-apoptotic effect on ALL cells treated with Adriamycin, leading to an increase in the expression of the Bcl-2 gene and an enhanced Bcl-2/Bax ratio [86]. A different study indicated that cell-to-cell contact between AML cells and MSCs promotes resistance against mitoxantrone in AML through c-Myc upregulation, preventing cancer cells from undergoing apoptosis [87]. Lastly, Carter et al. found that the interaction between AML cells and MSCs regulated by the ARC protein induces the expression of IL-1β in AML cells, which leads to resistance against cytarabine [88].

### 3.2. Anti-Tumorigenic Effects of MSCs in Leukaemia

MSCs can also have anti-tumorigenic effects through various mechanisms. One of the widely accepted mechanisms is tumour cell cycle arrest. Human umbilical-cord-derived MSCs (hUC-MSCs) were found to inhibit the proliferation of K562 cells by inducing an arrest in the G0/G1 phase through the secretion of IL-6 and IL-8 and involving the Notch signalling pathway [89,90,91]. Similarly, Zhu et al. investigated the impact of MSCs derived from human adipose tissue on the K562 CML cell line and found that these MSCs impeded tumour cell proliferation by inducing cell cycle arrest through dickkopf-related protein 1 (DKK1) secretion [92]. In another study, Wei et al. co-cultured K562 CML cells with MSCs derived from the BM of leukaemia patients and observed a decrease in the number of leukaemia cells in the S phase and an increase in the number of cells in the G0–G1 phase via the PI3K-Akt-Bad signalling pathway [93]. Han et al. found that BMSCs from both healthy individuals and those with CML increased the anti-apoptotic ability of cancer cells through regulation of Bcl-2, Bax, caspase-3, and activation of the Wnt signalling pathway in CML patients [94]. 

MSCs have been found to inhibit the growth of CML cells, likely through the production of IFN-α [95] or cytokine-induced neutrophil chemoattractant-1 (CINC-1) and tissue inhibitor of metalloproteinases-1 (TIMP-1) [96].

Notably, it has been suggested that MSCs play a dual role in leukaemia due to the heterogeneity of both leukaemia and MSCs [97]. The density of MSCs in the culture significantly impacts their proliferation rate, secreted factors, and morphology. The anti-tumorigenic effects of MSCs on solid cancers have been associated with a lower number of MSCs, whereas the pro-tumorigenic effect is observed with a higher number of MSCs [98]. However, due to a lack of data, this association has not been studied for leukaemia.

### 3.3. Changes in MSCs in Leukaemia

MSCs are believed to contribute to leukaemogenesis through genetic changes. For example, MLL-AF4 is a fusion gene commonly found in infant B-acute lymphoblastic leukaemia (B-ALL). A study showed that the MLL-ENL, MLL-AF10, or MLL-AF9 fusion genes were absent in the BMSCs of childhood leukaemia but that MLL-AF4 was detected and expressed in BMSCs from all cases of MLL-AF4+ B-ALL, indicating its possible tumour-associated role [99]. Additionally, MSCs from patients with childhood B cell precursor ALL have been found to exhibit various chromosomal translocations, including E2A-PBX1, TEL-AML1, or MLL rearrangement [100]. Another study found that BMSCs were cytogenetically abnormal in 54% of AML patients [101].

MSCs isolated from leukaemia patients exhibit functional abnormalities. The growth characteristics and cell proliferation of the BMSCs obtained from patients with CML were indistinguishable from those of the control group. However, the expression of integrin mRNA was found to be significantly higher in the former group, supporting CML cells and shielding them from the effects of immune surveillance [102].

MSCs in leukaemia are known to have immunomodulatory defects. MSCs from patients with ALL could inhibit T-lymphocyte proliferation stimulated in a mixed-lymphocyte reaction in a dose-dependent manner through TGFβ1 and HGF. In contrast, the MSCs from AML patients failed to suppress T cell proliferation [103]. A separate study showed that the MSCs from CML patients exhibited a reduced inhibitory effect on T cell activation and proliferation. The immune response type and severity vary according to the pathophysiological features of different kinds of leukaemia. The BMSCs in AML patients produce much lower levels of SCF, TNF-α, monocyte chemoattractant protein-1 (MCP-1), GM-CSF, and IL-6 compared to normal BMSCs from healthy controls. Lower levels of MCP-1 preclude the anti-tumour activity of MSCs and lower levels of GM-CSF suggest that the leukaemic BMME results in a diminished ability to support normal haematopoiesis [104]. The increased level of IL-8 produced by MSCs in AML leads to leukaemogenesis [105]. Furthermore, the elevated level of indoleamine 2,3-dioxygenase (IDO) secreted by MSCs in leukaemia leads to the apoptosis of T cells, thereby reducing the anti-leukaemia immune response [106].

BM abnormalities contribute to the development and expansion of MDS or leukaemia clones and, in turn, MSCs or stromal cells promote clonal expansion and disease progression (Figure 3). Reciprocal heterotypic signalling between MSCs and disease-propagating haematopoietic cells within the BM might be required for disease initiation/progression. 

## 4. Potential Use of MSCs in Therapies for Blood Cancers

MDSs are treated with hypomethylating agents (HA), such as decitabine and azacitidine (AZA), immunomodulatory drugs, and chemotherapy, depending on the subtype of the MDS and the severity of symptoms [107].

MSCs are a promising treatment option for MDSs and have the potential to improve their functions in this context. This is due to the remarkable immunomodulatory properties of MSCs, their ability to arrest the cell cycle, and their potential as delivery vehicles. Moreover, these cells are readily obtainable and can be extensively expanded in vitro [108]. Also, MSCs have a natural inclination to migrate towards tumour environments and injured tissues [109].

AZA treatment was found to restore the aberrant hypermethylation pattern of MDS-BMSCs, leading to an increased proliferation potential and osteogenic capacity for the cells and an improved ability to support HSPCs for in vivo engraftment [43]. Additionally, MSCs treated with AZA cause a remarkable reduction in the increased level of IL-6 in MDS-MSCs, ameliorating the inflammatory environment induced by MSCs [110]. Another study revealed that AZA-treated MDS-MSCs improved haematopoiesis in HR-MDS and reversed MDS-MSCs’ proliferation and osteogenic differentiation capacities by normalising the BMME disrupted by MSCs [41]. 

Decitabine treatment reduces the ability of MDS-MSCs to induce the differentiation of T cells into Treg cells. This reduction is related to decreased programmed death-ligand 1 (PD-L1) expression in decitabine-treated MDS-MSCs. This change was related to a decrease in cyclin-dependent kinase inhibitor 1A (CDKN1A) expression, which suggests that the senescence of MDS-MSCs was improved [111]. 

Furthermore, the administration of α-lipoic acid (ALA) can reduce reactive oxygen species (ROS) levels in MSCs and decrease the intracellular iron content that results from ineffective erythropoiesis, which may be a useful factor against MDS [112]. When BMSCs are co-cultured with MDS-derived cells, menatetrenone has been observed to enhance apoptosis, indicating that it can aid BMSCs in supporting haematopoiesis and improve cytopenia in MDSs [113]. The ability of patient-derived MSCs to undergo osteogenic differentiation was restored through the blockade of TGFβ signalling with SD-208.

Studies have shown that PI3K inhibitors, such as PIK-90 and PI-103, can reverse the MSC-induced fludarabine resistance in CLL cells [114]. CXCL12, which is crucial for enhancing the homing of CXCR4-expressing HSCs into BM, is secreted at higher levels in MDS-MSCs [115], possibly accounting for BM hypercellularity. Treatment with lenalidomide that can regulate the expression of chemokines in MSCs has been demonstrated to decrease C-X-C motif chemokine ligand 12 (CXCL12) secretion by MDS-BMSCs, supporting normal HSPCs [38]. 

The Notch signalling pathway is another therapeutic target. Studies have shown that using anti-notch molecule-neutralising antibodies can decrease B-ALL cell survival. Furthermore, blocking the Notch pathway entirely using GSI XII can cause an increase in the number of apoptotic B-ALL cells co-cultured with MSCs [116]. Finally, the use of hEGR1 blockers, such as E4031, has been proposed as a potential strategy to overcome chemotherapy resistance in ALL cells that are induced by MSCs through hEGR1 channels. A study conducted on NOD/SCID mice that were engrafted with ALL cells demonstrated that following treatment with specific hERG1 channel blockers, a decrease in leukaemic infiltration was observed and the survival rate of the mice was significantly higher [117]. Further research is warranted to elucidate the mechanisms underlying the effects of these therapeutic strategies and assess their clinical efficacy.

It has been discovered that transferring the hIFN-γ gene to MSCs can reduce the proliferation of leukaemia cells. Co-culturing genetically modified MSCs with K562 cells led to an increase in apoptotic K562 cells compared to the negative control [118]. Another study used MSCs genetically engineered to release an anti-CD33–anti-CD3 bispecific antibody for immunotherapy in AML [119]. Finally, it has been shown that engineered MSCs are effective delivery systems for gene therapy in treating leukaemia. When these cells were injected into mice with CML, the mice showed tumour regression and improved survival rates due to the in vivo interferon production by the engineered MSCs [120].

MSCs have been shown to have the potential to deliver chemotherapeutic drugs. A study demonstrated that when human MSCs were loaded with PTX, they were able to effectively inhibit the growth of leukaemia cells and angiogenesis in vitro. The study also observed that when MSCs were primed with PTX, it had a positive impact on the survival of leukaemia-bearing mice in vivo [121]. Similarly, exosomes derived from BMSCs delivered miR-222-3p through the targeting of IRF2, inhibiting cell proliferation and promoting cell apoptosis in the AML cell line [122]. One study showed that MSCs carrying levamisole (sLipo leva) could produce an anti-leukaemia effect in mice affected with leukaemia through targeting macrophages [14]. These discoveries highlight the potential of MSCs as effective delivery systems in leukaemia gene therapy.

A recent study has demonstrated that intra-BM treatment that includes the direct injection of donor MSCs into the tibia of a leukaemia-bearing mouse can restore the BMME. Additionally, intra-BM treatment with MSCs decreased tumour burden and prolonged the survival of leukaemia-bearing mice. The study also found that donor MSC treatment restored the function of host MSCs and reprogramed the host macrophages into an arginase1-positive phenotype with tissue-repair features. Based on these findings, the study concluded that donor MSCs could reprogram host macrophages for the restoration of the BMME and inhibition of leukaemia development [123]. Table 3 presents a compilation of all studies that have explored the impact of MSCs on leukaemia by suppressing tumour growth and the studies that targeted MSCs or used them as drug delivery tools.

There are approximately 16 clinical trials registered on clinicaltrial.gov that aim to utilise MSCs in the treatment of haematologic malignancies. Out of these trials, seven involve the use of engineered MSCs as carriers for therapeutic cytokines or oncolytic viruses, while one trial is specifically designed to assess the safety and efficacy of MSC-derived exosomes containing KrasG12D siRNA (iExosomes) [124]. A phase I/II randomised study registered with NCT04565665 explains that cord-blood-derived MSCs, which are a form of tissue-derived MSC, have been adopted for the treatment of haematopoietic and lymphoid cell neoplasms, and the trial is currently recruiting participants. Likewise, the NCT03184935 clinical phase I/II study makes use of cord-blood-derived MSCs for the treatment of myelodysplastic syndrome patients. The clinical trials currently using tissue-derived-MSC-based therapies for the treatment of haematologic malignancies are listed in Table 4.

## 5. Conclusions and Future Perspectives

The anti-tumorigenic effects of MSCs are primarily attributed to their ability to inhibit the proliferation of leukaemia cells. The precise mechanisms or molecules involved in this process remain unclear. To leverage this antitumour activity for clinical applications in the future, it is imperative to consider other factors. MSCs possess certain beneficial properties, including the potential to be used as delivery vehicles and the ability to inhibit vascular growth and arrest the cell cycle. MSCs exhibit several valuable properties, including their potential to serve as delivery vehicles, their ability to inhibit vascular growth, and their capacity to arrest the cell cycle. However, they also possess certain unfavourable characteristics, such as their tendency to promote tumour growth by suppressing apoptosis, supporting tumour vasculature, and modulating the immune response of cancer cells. Furthermore, they contribute to the protection of cancer cells from drug-induced apoptosis, thereby causing chemo-resistance, which poses a significant obstacle to their use as a therapeutic agent in leukaemia.

In addition, despite the broad acceptance of the dual role of MSCs in leukaemia, a solid principle that explains the anti-tumorigenic and pro-tumorigenic effects of MSCs is imperative. This principle will provide greater clarity on the impact of MSCs in leukaemia treatment and aid in the development of more effective therapeutic approaches.

Developing anti-leukaemia therapies using MSCs that target individual pathways, such as CXCR4/CXCL12, Notch, and Wnt, may be feasible. Specifically, developing molecules that augment anti-tumorigenic effects or diminish pro-tumorigenic effects would be a promising approach for advanced leukaemia therapies. Further investigation of the anti-tumorigenic effects of MSCs is warranted to develop an effective and safe treatment for leukaemia. This could involve the development of engineered or genetically modified MSCs that are more efficacious than the heterogeneous and unstable naïve MSCs.

MSC-based clinical results have exhibited a wide range of variation, likely attributable to the absence of standardised experimental methods, specific cell surface markers to identify subsets of MSCs, and the heterogeneous characteristics of MSCs that are easily influenced by the surrounding environment. Thus, further investigation is imperative to advance the development of MSCs for cancer treatment. 

From the various underlying mechanisms proposed and summarised herein, it may be conceivable to formulate MSC-based anticancer therapies by selectively targeting individual pathways. Specifically, creating molecules capable of augmenting the antitumorigenic effects or attenuating the pro-tumorigenic effects holds considerable promise for advancing therapeutic strategies. Elaborate studies are requisite to surmount limitations such as the heterogeneous nature of MSCs and the absence of standardised methodologies. Further investigation pertaining to the antitumor effects of MSCs ought to be conducted to devise secure and efficacious treatments for haematologic malignancies. Developing engineered or genetically modified MSCs may serve as a propitious approach, as they exhibit enhanced safety and efficiency in contrast to the unstable and heterogeneous naïve MSCs. As numerous researchers persevere in overcoming the limitations of and refining MSC-based cellular therapies targeting haematologic malignancies, optimism exists that successful therapeutic interventions will be realised. Until such time, it is indispensable that we adopt a cautious approach to MSC-based cellular therapies, considering the limitations and unexpected outcomes delineated.

## Figures and Tables

**Figure 1 biomedicines-12-01677-f001:**
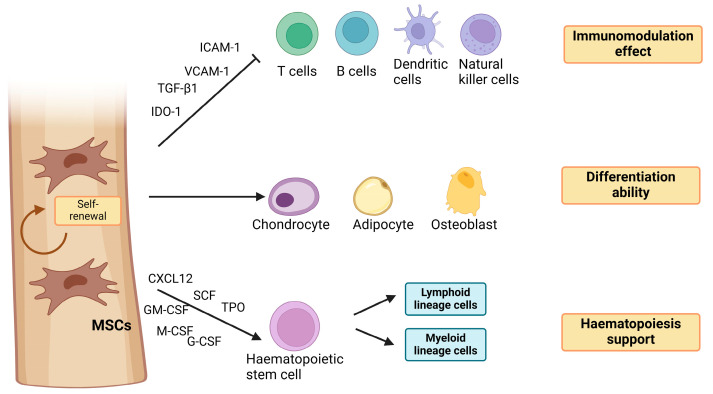
Functions of MSCs. MSCs can differentiate into different types of cells, support haematopoiesis, self-renew, and exert an immune-modulation effect. ICAM: Intercellular adhesion molecule 1, VCAM: Vascular cell adhesion molecule1, TGF-β1: Transforming growth factor beta-1, IDO: Indoleamine-pyrrole 2,3-dioxygenase, CXCL12: C-X-C motif chemokine ligand 12, SCF: Stem cell factor, TPO: Thrombopoietin, GM-CSF: Granulocyte-macrophage colony-stimulating factor, M-CSF: Macrophage colony-stimulating factor, G-CSF: Granulocyte-colony stimulating factor. (Created with BioRender.com under NTU licence).

**Figure 2 biomedicines-12-01677-f002:**
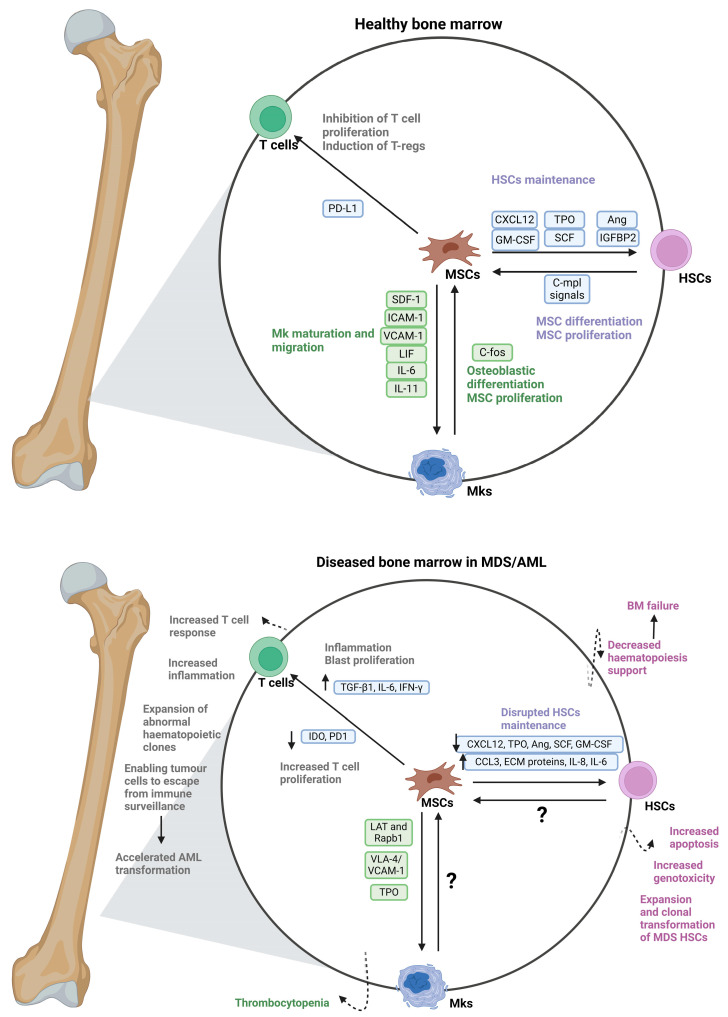
Effect of impaired MSCs in MDS compared to normal. MDS-MSCs exhibit inflammatory alterations, dysregulation of immune modulation, defective differentiation potential, and a decrease in proliferation and growth, leading to an increase in apoptosis and genotoxicity, together with the expansion and clonal evolution of HSCPs in MDS and impaired support of normal HSPCs. (MSCs: Mesenchymal stem cells, HSCs: Haematopoietic stem cells, Mks: Megakaryocytes, PD-L1: Programmed death ligand-1, TGF-β1: Transforming growth factor beta, IL-6: Interleukin-6, IFN-γ: Interferon-gamma, IDO: Indoleamine 2,3-dioxygenase, CXCL12: C-X-C motif chemokine ligand 12, Ang: Angiopoietin, GM-CSF: Granulocyte-macrophage colony-stimulating factor, CCL3: C-C motif chemokine ligand 3, ECM: Extracellular matrix, IL-8: Interleukin-8, (Created with BioRender.com under NTU licence).

**Figure 3 biomedicines-12-01677-f003:**
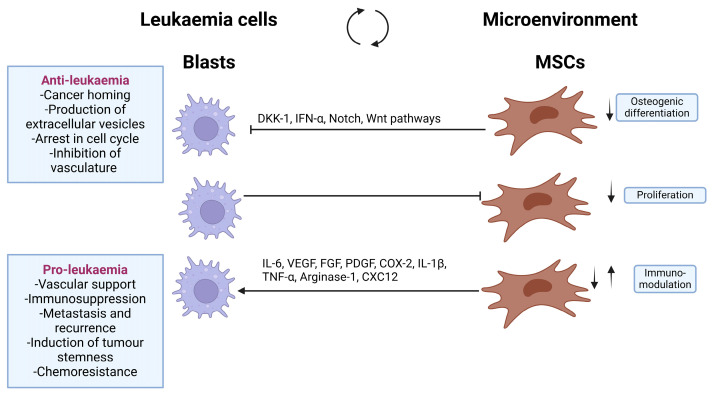
Development and progression of MDS and the biphasic role of MSCs in leukaemia. MSCs have anti-tumorigenic and pro-tumorigenic effects on leukaemia cells. (DKK1: Dickkopf-related protein 1, IFN-α: Interferon-alpha, IL-6: Interleukin-6, VEGF: Vascular endothelial growth factor, FGF: Fibroblast growth factor, PDGF: Platelet-derived growth factor, COX-2: Cyclooxygenase 2, IL-1β: Interleukin-1beta, TNF-α: Tumour necrosis factor alpha, CXCL12: C-X-C motif chemokine 12). (Created with BioRender.com under NTU licence).

**Table 1 biomedicines-12-01677-t001:** MDS subtypes according to WHO 2022 classification.

MDSs with Genetic Abnormalities	Blasts Present in BM and PB
MDS with low blast level and 5q-del	<5% BM and <2% PB
MDS with low blasts level and mutated SF3B1	<5% BM and <2% PB
MDS with biallelic TP53 inactivation	<20% BM and PB
MDS morphological dysregulations	
MDS with low blasts (MDS-LB)	
MDS, hypoplastic (MDS-h)	
MDS with increased blasts (MDS-IB)	
MDS-IB1	5–9% BM or 2–4% PB
MDS-IB2	10–19% BM; or 2–19% PB
MDS with fibrosis	5–19% BM; or 2–19% PB

**Table 2 biomedicines-12-01677-t002:** MDS subtype scores according to IPSS classification.

Total IPSS Score	Risk Groups	Median Survival	How Long Does It Take to Develop AML?
0	Low risk	5.7 years	9.4 years
0.5–1	INT-1 risk	3.5 years	3.3 years
1.5–2	INT-2 risk	1.2 years	1.1 years
≥2.5	High risk	0.4 years	0.2 years

**Table 3 biomedicines-12-01677-t003:** Studies of MSCs suppressing leukaemia growth and studies targeting/using MSCs as a delivery tool for the treatment of AML and MDS.

MSCs Type	Tumour Type	Effect	Reference
Mouse BMSCs	ALL (P388) and B-lymphoma (A20)	Leukaemia/lymphoma cell growth inhibition	[19]
Human BMSCs	CML (BV173) and T-ALL (Jurkat)	Inhibition of cancer cell proliferation	[66]
UC-MSCs	CML (K562)	Anti-proliferative effect on leukaemia cells	[44]
Human AT-MSCs	AML (HL-60) and CML (K562)	Leukaemic cell growth inhibition	[92]
Human BMSCs	CML (patient’s cells)	CML cell growth inhibition	[95]
Human BMSCs	CML (K562)	CML cell growth inhibition	[96]
Human UC-MSCs	CML (K562)	K562 cell growth inhibition	[89]
Human adipose tissue MSCs	CML (K562) and AML HL-60)	Inhibition of cancer cell proliferation	[92]
Leukaemia patient’s BMSCs	CML (K562)	Inhibition of apoptosis and leukaemia cell growth	[93]
Human UC-MSCs	AML (HL-60) and CML (K562)	Inhibition of cancer cell proliferation	[90]
Human BMSCs and CML patient’s BMSCs	CML (K562 and patient’s cells)	Enhancement of the anti-apoptotic capacity of cancer cells	[94]
Human UC-MSCs	T-ALL (Jurkat cell line)	Jurkat cell growth inhibition	[91]
Patient-derived BMSCs	MDS	Improvement of proliferation and osteogenic capacity of BMSCs with their increased support of HSPCs	[43]
Patient-derived BMSCs	MDS	Improvement of the negative impact of MSCs on haematopoiesis by the support of MSCs on healthy HSPC expansion	[84]
Patient-derived BMSCs	MDS	Improvement of the inflammatory environment through AZA	[110]
Patient-derived BMSCs	MDS	Improvement of haematopoiesis through AZA	[41]
Patient-derived BMSCs	MDS	Improvement of BMSCs’ capacity for supporting normal HSPCs through lenalidomide	[38]
Patient-derived BMSCs	MDS	Reducing the ability of MSCs to induce the differentiation of T cells into Tregs and improvement of MDS-MSC senescence through decitabine	[111]
Patient-derived BMSCs	MDS	Suppression of the adhesion of leukaemic cells to the stroma	[115]
Human BMSCs	MDS	Decrease in autophagy	[112]
Human BMSCs	MDS	Improvement of the proliferation activity of MSCs and BMSC support in haematopoiesis	[113]
Patient-derived BMSCs	MDS	Restoration of the osteogenic differentiation of MSCs	[39]
BMSCs	CML	Regression of tumour and improvement of survival rates	[120]
Mice BMSCs	T-ALL	Decrease in tumour burden and improvement of survival rate	[123]
Human BMSCs	CML (K562)	Reduction in the proliferation of CML cells and induction of apoptosis	[118]
Human BMSCs	AML	Increase in the survival of MSCs	[119]
Human BMSCs	Human T-cell ALL (MOLT-4) and mouse CLL (L1210)	Leukaemic cell growth inhibition and inhibition of angiogenesis	[121]
Human UC-MSCs	CML (K562)	Leukaemic cell growth inhibition	[44]
Rat BMSCs	ALL (Ball-1) and K562 (CML)	Leukaemic cell growth inhibition	[14]
Human BMSCs	AML (THP-1)	AML cell growth inhibition and induction of AML apoptosis	[122]

**Table 4 biomedicines-12-01677-t004:** Clinical studies utilising MSC-based approaches for the treatment of haematologic malignancies (obtained from clinicaltrials.gov).

NCT No.	Phase	Interventions	Treatment	Cancer Applications
NCT04565665	I/II	Cord blood MSCs	MSCs IV followed by a second fusion of MSCs within 7 days of the first one.	Haematopoietic and lymphoid cell neoplasm
NCT03184935	I/II	Cord blood MSCs	Allogeneic umbilical cord MSCs and decitabine (20 mg/m^2^)	Myelodysplastic syndromes
NCT02181478	I	MSCs	Reduced-intensity conditioning with cyclophosphamide, fludarabine (with total body irradiation), or fludarabine and melphalan followed by co-transplantation of intra-osseous umbilical cord blood and MSCs.	Haematologic malignancies
NCT01624701	I/II	Bone marrow MSCs	Clinically ex vivo expanded cord blood cells are comprised of stem cell factor, Flt3 ligand, thrombopoietin, IGFBP2, and MSC co-culture.	Expanding umbilical-cord-blood-derived blood stem cells for treating leukaemia, lymphoma, and myeloma
NCT01092026	I/II	Cord blood transplantation + MSCs	Umbilical cord blood haematopoietic stem cell transplantation co-infused with third-party MSCs	Haematologic malignancies
NCT01045382	II	Haematopoietic stem cells + MSCs	1.5–3.0 × 10 × 10^6^ MSC/Kg with fludarabine and 2 Gy total body irradiation followed by HLA-matched PBSC	Leukaemia, lymphoma, and myeloma
NCT01129739	II	Cord blood MSCs	1 × 10^6^ MSC/kg, intravenous	Myelodysplastic syndromes
NCT05672420	Ib/II	Umbilical-cord-derived MSCs	RP2D, intravenous	Haematologic malignancies

## Data Availability

No new data were created or analysed in this study.

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
