# Peer review of "Mesenchymal Stem Cells in Myelodysplastic Syndromes and Leukaemia"

_biomedicines, 2024, doi:10.3390/biomedicines12081677_

Round 1

Reviewer 1 Report

Comments and Suggestions for Authors

Dear Authors!

Thank you for the opportunity to review your manuscript.

This is a classical review about the role of  the mesenchymal stem cells in myelodysplastic syndromes and leukaemia microenvironment.

In this review Authors cited more than 100 manuscripts and provided the know data about these conditions.

The manuscript is well written and well-orf=ganized. It contains from multiple subsection, related to every specific problem

The idea of manuscript is actual, because these disease are severe and Authors provided the comprehensive view about these conditions.

The manuscript is too long and it will be difficult for reading.

Authos provided several figures, please indicated if they are original or adopted? Every figure required to full names of abbreviations in the footnotes to every figure.

Author cited relevant manuscripts and list of them also too big.

I have no principal concerns about the quality of manuscript, including organization and Englishe quality, but I can recommend to make it more concise to readers.

Author Response

Comment: The manuscript is too long and it will be difficult for reading.

Authors provided several figures, please indicated if they are original or adopted? Every figure required to full names of abbreviations in the footnotes to every figure.

Author cited relevant manuscripts and list of them also too big.

I have no principal concerns about the quality of manuscript, including organization and English quality, but I can recommend to make it more concise to readers.

Response:

We would like to thank the reviewer for their constructive comments.

As recommended and to make it concise to readers:

-We have now shortened the review (shortened long parts and removed some parts from MDS classifications and parts about other blood cancers, lymphoma and multiple myeloma).

-We also reduced the reference list to 121 references.

-We also updated the figures, so we have three figures (figures 3 and 4 were combined). We added that figures were made using BioRender software under NTU licence.

Reviewer 2 Report

Comments and Suggestions for Authors

The article entitled “Mesenchymal Stem Cells in Myelodysplastic Syndromes and

Leukaemia Microenvironment” addresses a broad topic about the numerous mechanisms causing the myelodispastic syndromes and leukemias. Hovewer, this review do not address an important biological mechanism such as the epigenetic  (Seishi Ogawa Blood 2019; Bhagat TD et al, Cancer Res. 2017; Nguyen CT et al, Cancer Res 2002). Therefore, I think that the authors should add a paragraph about this argument. In addition, I suggest changing the title as follows “Mesenchymal Stem Cells in Myelodisplastic Syndromes and Leukemias” and deleting the paragraph n. 4  “Role of MSCs in Lymphoma and Multiple Myeloma Pathogenesis”. Therefore, I think that this article is not suitable for publication in its current version.

Author Response

Comment 1: The article entitled “Mesenchymal Stem Cells in Myelodysplastic Syndromes and Leukaemia Microenvironment” addresses a broad topic about the numerous mechanisms causing the myelodispastic syndromes and leukemias. Hovewer, this review do not address an important biological mechanism such as the epigenetic  (Seishi Ogawa Blood 2019; Bhagat TD et al, Cancer Res. 2017; Nguyen CT et al, Cancer Res 2002). Therefore, I think that the authors should add a paragraph about this argument.

 Response 1: We thank the reviewer for the constructive comments. As recommended by the reviewer, we added a paragraph about the epigenetic role and cited these references.

Comment 2: In addition, I suggest changing the title as follows “Mesenchymal Stem Cells in Myelodisplastic Syndromes and Leukemias” and deleting the paragraph n. 4  “Role of MSCs in Lymphoma and Multiple Myeloma Pathogenesis”.

Response 2: As recommended by the reviewer, we changed the title and deleted paragraph n. 4, “Role of MSCs in Lymphoma and Multiple Myeloma Pathogenesis”.

Round 2

Reviewer 2 Report

Comments and Suggestions for Authors

In the revised version of the article entitled “Mesenchimal Stem Cells in Myelodisplastic Syndromes and Leukaemia Microenvironment” the authors modified the manuscript point by point in according to the comments of the reviewer. Therefore, I think that this article is suitable for publication in its current revised version.